# Agreement between Family Members and the Physician’s View in the ICU Environment: Personal Experience as a Factor Influencing Attitudes towards Corresponding Hypothetical Situations

**DOI:** 10.3390/healthcare11030345

**Published:** 2023-01-25

**Authors:** Paraskevi Stamou, Dimitrios Tsartsalis, Georgios Papathanakos, Elena Dragioti, Mary Gouva, Vasilios Koulouras

**Affiliations:** 1Intensive Care Unit, University Hospital of Ioannina, University of Ioannina, 45500 Ioannina, Greece; 2Department of Emergency Medicine, “Hippokration” Hospital, 11527 Athens, Greece; 3Pain and Rehabilitation Centre, Department of Health, Medicine and Caring Sciences, Linköping University, SE-581 83 Linköping, Sweden; 4Laboratory of Psychology of Patients, Families & Health Professionals, Department of Nursing, School of Health Sciences, University of Ioannina, 45500 Ioannina, Greece

**Keywords:** intensive care unit, resilience, realism, family, withdrawal decision

## Abstract

Background: It is not known whether intensive care unit (ICU) patients’ family members realistically assess patients’ health status. Objectives: The aim was to investigate the agreement between family and intensivists’ assessment concerning changes in patient health, focusing on family members’ resilience and their perceptions of decision making. Methods: For each ICU patient, withdrawal criteria were assessed by intensivists while family members assessed the patient’s health development and completed the Connor–Davidson Resilience Scale and the Self-Compassion Scale. Six months after ICU discharge, follow-up contact was established, and family members gave their responses to two hypothetical scenarios. Results: 162 ICU patients and 189 family members were recruited. Intensivists’ decisions about whether a patient met the withdrawal criteria had 75,9% accuracy for prediction of survival. Families’ assessments were statistically independent of intensivists’ opinions, and resilience had a significant positive effect on the probability of agreement with intensivists. Six months after discharge, family members whose relatives were still alive were significantly more likely to consider that the family or patient themselves should be involved in decision-making. Conclusions: Resilience is related to an enhanced probability of agreement of the family with intensivists’ perceptions of patients’ health progression. Family attitudes in hypothetical scenarios were found to be significantly affected by the patient’s actual health progression.

## 1. Introduction

Intensive care units (ICUs) are for patients with medical conditions that imminently threaten their survival. Being hospitalized in ICU means that the patient’s physical health has suffered excessive and possibly irreversible damage [1]. Additionally, the generally unexpected admission of a patient to the ICU can be particularly frightening and distressing for their loved ones [2]. Higher levels of anxiety, depression, and stress during admission are commonly reported in the literature [3], while post-traumatic stress disorder and complicated grief occur after discharge [4]. Nonetheless, family members are seen as an integral part of the healthcare process and the need for good collaboration should always be considered. 

The most important decision made during an ICU stay is often whether to use life-support devices to prolong life, or to discontinue life support and place more emphasis on comfort measures, given that further intervention is futile [5]. To make such a decision as objectively as possible, intensivists have been extensively and thoroughly trained to ensure that their judgment is based on globally recognized health indicators that objectively determine the patient’s clinical picture.

Under these circumstances, it is common for the patients not to be able to express their wishes to the medical team [6]. The role of family members is very important, as they are called upon not to express their own opinion on whether the patient’s life should be further mechanically assisted, but the patient’s own view, based on their perception of the patient’s personality and character, or after relevant discussions with the patient prior to admission to the ICU. It is not clearly known whether patients’ relatives realistically assess the patient’s status of health [6,7]. In fact, an unrealistic perception of the patient’s condition on the part of loved ones leads to tension and feelings of unease about the decision to withdraw life support, reflected in overall satisfaction with healthcare system performance [3,8,9]. 

Moreover, little is known about the factors that influence the patient’s family members when asked to make end-of-life decisions on the patient’s behalf. In this context, the concepts of mental resilience and self-compassion have been implicated as potential factors in the psychological well-being of families in the ICU, including the subject’s ability to succeed despite the adversities they face in life [10] and the subject’s ability to have a warm, caring, empathetic, and non-judgmental orientation towards the self at times of suffering and failure [11,12]. Specifically, our group found that self-compassion and mental resilience were highlighted as the two psychological traits that explain the overall psychological distress experienced by attendants in the ICU environment [13]. 

It is essential to examine the factors influencing the realistic view formed by the patient’s relatives, because these directly relate to the quality of communication with medical staff and the overall experience in the ICU. Therefore, the question arises whether these two characteristics are also related to the attendant’s increased ability to assess realistically the patient’s state of health and to agree with the intensivist’s opinion. 

A primary aim of the present study was to fill this research gap by examining whether demographic variables, resilience, and self-compassion also influence family members’ realistic view of the patients’ health. Then, we considered the post-ICU attitudes of family members, and examined whether realistic attitudes during the experienced situation were related to the belief that the family should participate in decision-making in other hypothetical situations, and whether the evolution of the patient’s health played an important role. In doing so, we also assessed the validity of hypothetical scenarios as tools for identifying attitudes and perceptions, and their usefulness as policy-making tools.

## 2. Materials and Methods

### 2.1. Participants and Study Design

A cross-sectional study was conducted using a quantitative methodology, to evaluate psychological impact on relatives of critically ill patients. The data were collected in two time periods from 2019 to 2021; the first took place during the patient’s hospitalization in the ICU of our tertiary university hospital, while the second took place six months after the patient’s ICU discharge. 

A total of 162 patients and their 189 family members, i.e., spouse, child, parent, or other, were recruited and agreed to participated in the study. Family members of patients with elective postoperative admission or brain death or who died within 1 week after admission were excluded from the study. Oral informed consent was obtained from family members and the study was approved by the Ethics Committee of the University Hospital of Ioannina.

Within the first two days after patient admission, the Glasgow coma score (GSC), the acute physiology, age and chronic health evaluation score (APACHE), and the simplified acute physiology score (SAPS) were assessed by ICU physicians. When the medical status and the prognosis of the patient were clarified, the director of the department together with the 3 most experienced intensivists completed a brief screening questionnaire describing whether the patient met any of eight criteria for withdrawal of treatment, along with a single direct question as to whether the patient would eventually survive. Then, 7–10 days after patient’s admission to the ICU and in the knowledge of the intensivists’s assessment of the patient’s health status, the relatives were asked to complete a multiple-choice questionnaire. For each family member, gender, age, type of relationship, and their assessment of the health progression of their relative were recorded on a five-point Likert scale (1: hopeless to 5: hopeful). 

Additionally, each family member completed the Connor–Davidson resilience scale (CD-RISC) [14] and the self-compassion scale (SCS) [12]. The CD-RISC consists of 25 items that are answered on a 5-point frequency scale (0 to 4). CD-RISC’s total score ranges from 0 to 100, with higher scores indicating greater perceived resilience [14]. The SCS consists of 26 questions answered on a 5-point frequency scale (1 to 5), and the total score is calculated as the overall mean after 13 of the score values are reversed [12]. The total score reflects self-compassion as defined as a dynamic balance between compassionate and uncompassionate ways in which individuals respond emotionally to pain and failure, cognitively understand their predicament, and pay attention to suffering [12]. 

Six months after each patient’s ICU discharge, telephone follow-up contact was established with 153/189 (81%) of participants, all of whom had close contacts with a patient. During this telephone interview, family members answered four questions about two hypothetical clinical scenarios: one with a conscious and competent patient being able to comprehend his actual state of health, and one with an unconscious patient who cannot participate in medical decisions that affecting him. In both scenarios, the first question assessed whether the patient’s family should be involved in the decision to withdraw life support measures, while the second question aimed to capture the family member’s opinion about who should be responsible for making the decision. The two scenarios were taken from a previous study [15] and translated into Greek with minor changes in the responses, allowing independent selection of all those involved in the decision-making process. 

### 2.2. Statistical Analysis

The Chi-square test of independence was applied to evaluate whether two nominal or ordinal variables were statistically independent. Analysis of variance was applied to quantify the differences between more than two groups, while Tukey’s b test was employed to highlight the homogeneous groups. To elucidate the similarities between respondents’ answers in the hypothetical scenarios, the distances between pairs of binary variables were computed using the Dice coefficient of similarity (known also as the Czekanowski or Sorensen measure) [16]. Then, a hierarchical cluster analysis was applied to provide an indicative grouping of similar responses. A logistic regression model was applied to test whether demographic factors, resilience, and self-compassion affected agreement between family members’ and intensivists’ assessments, and a second logistic model was applied in order to test whether agreement in assessment and the progression of patient’s health affected respondents’ perceptions in analogous scenarios.

A two-sided level of significance of 0.05 was set for all statistical tests. The data were analyzed using SPSS statistical package (version 21) and R statistical language [17].

## 3. Results

### 3.1. Sample Characteristics 

The demographic sample characteristics for family members and patients are presented in Table 1. The mean age of family members was 46.5 (SD 11.4 years) and the corresponding figure for the patients was 64.4 (SD 17.2 years). Among the family members, 111 (58.7%) were women; the corresponding figure for the patients was 52 (32.1%). 

### 3.2. Intensivists Criteria for Withdrawing Life-Sustaining Treatment

Among the 162 patients admitted to the ICU, 46 (28.4%) met the intensivists’ criteria for treatment withdrawal. Lack of future quality of life and futility of treatment were the dominant clinical assessments. Meanwhile, hospital costs were not regarded as a withdrawal criterion for any patient, while age was considered a criterion for six patients (M = 78.7 years, SD = 3.8) (Table 2).

The 46 patients that met the withdrawal criteria were significantly older (73.3 ± 11.5 vs. 60.9 ± 17.8, *p* < 0.001) than the others, they were characterized by significantly higher SAPS scores (54.7 ± 13.8 vs. 39.1 ± 12.7, *p* < 0.001) and APACHE scores (21.5 ± 5.7 vs. 15.9 ± 5.8, *p* < 0.001), and significantly lower GCS scores (7.0 ± 3.0 vs. 10.1 ± 3.5, *p* < 0.001). During ICU hospitalization, 25/46 passed away, with the remaining 21 patients at ICU discharge presenting severe disability regarding feeding (nasogastric tube or gastrostomy), breathing (tracheostomy), or mobility (hemiplegia, tetraplegia, critical care myopathy). Six months after ICU hospitalization, nine patients were still alive with little (7) or moderate (2) health recovery, and unable to live autonomously. Overall, the accuracy of the physician’s classification (PAC) concerning patient’s survival was 75.9% during hospitalization, and 71% six months after hospitalization (Table 3).

### 3.3. Family Members’ Agreement with Intensivists Concerning Patient’s Health

About half of the 189 respondents (86, 45.5%) were overly optimistic about the patient’s health progress (Table 4). The subjective optimism expressed by the patients’ family members was statistically independent of the intensivist’s evaluation (c^2^(4) = 6.279, *p* = 0.179). The family members were divided into three categories according to their assessments, in comparison with the those of the intensivists. The first category comprised family members who did not expect a positive change in the patient’s health, while the doctors insisted on the continuation of life support (*N* = 38). The second group contained the family members who perceived the patient’s state of health in agreement with the intensivist’s perception (*N* = 108), and the third category included family members who expected a positive development in the patient’s health while the intensivists suggested withdrawal of life support (*N* = 36). Overall, 74 (40.7%) of the respondents were not in agreement with intensivists’ judgments about changes in the health of their relative (Table 4).

The mean resilience score for the total sample of family members was 70.8 (SD = 14.4) analogous to the general Greek population (MP = 70.2, SD = 11.4) [18]. The mean score for self-compassion was 3.3 (SD = 0.5), considered moderate (2.5 to 3.5) due to the lack of clinical norms or scores to suggest that an individual has high or low self-compassion [19].

Logistic regression was carried out to quantify the effects of the patients’ age and gender, family members’ age and gender, staying with the patient, resilience, and self-compassion in terms of the agreement between the attendant’s and intensivist’s assessment of the patient’s health. The logistic model was statistically significant (omnibus test of model coefficients: c2(8) = 19.432, *p* = 0.013), being able to predict correctly 66.7% of the observations (sensitivity 81.6%, specificity 45.1%, 2-log likelihood = −215.865, McFadden’s pseudo R squared = 0.083) (Table 5). 

The attendant’s resilience (B = 0.031, ExpB = 1.032, 95% C.I. 1.005–1.060, *p* = 0.022), had a significant effect on the probability of agreement concerning the patient’s health. Specifically, an additional score of one on the resilience scale corresponded to 1.032 times greater likelihood that the respondent agreed with the intensivist’s view of the patient’s health. In particular, those who agreed were characterized by a significantly higher resilience score (M_NA_ = 68.0 vs. M_AG_ = 73.0, t(172) = 2.402, *p* = 0.017). 

The effect of the patient’s age on realism was marginally rejected at the 0.05 level (B = −0.020, ExpB = 0.980, 95% C.I. 0.960–1.000, *p* = 0.050), suggesting a noteworthy but not significant effect. In this context, it is worth noting that the statistically significant difference between patient age in the two groups (M_NA_ = 68.0 vs. M_AG_ = 60.6, t(172) = 2.775, *p* = 0.006).

### 3.4. Family Participation in the Theoretical Scenarios, 6 Months after the ICU Experience

At the second sampling timeperiod, among the 153 family members that responded to the study, 13 were parents, 31 were spouses, 18 were brothers, and 91 were offspring. In both hypothetical scenarios, most of the respondents favored family participation in decision making (Scenario 1: 78, 51.0%, Scenario 2: 109, 71.2%) (Table 6). 

A logistic regression analysis was applied to test the effects of patient’s age, agreement with intensivists during hospitalization, and survival of the patient on the probability of considering family or patient responsible for the withdrawal decision. The logistic model was statistically significant (Omnibus Test of Model Coefficients: c2(5) = 12.888, *p* = 0.024), being able to predict 66.7% of the observations correctly (sensitivity 93.9%, specificity 18.2%, −2-log likelihood = −186.966, Nagelkerke pseudo R squared = 0.111) (Table 7).

A significant interaction between the patient’s age and his or her health condition was identified (Wald W = 6.873, df = 2, *p* = 0.032). Specifically, as the age of patients increased, respondents whose relatives were still alive and lived autonomously were significantly more likely to consider the patient himself or the family as those who should be involved in decision making. In contrast, in cases where the patient had died, as the age of their deceased relatives increased the respondents tended to hesitate to declare the family responsible for decision making in the theoretical scenarios (Figure 1).

## 4. Discussion

Withdrawal of life-sustaining therapy, while not a strictly documented procedure, is an ethically acceptable practice in western ICUs. For example, in the Ethicus-2 study, a prospective observational study of 199 ICUs in 36 countries involving 87,951 patients who were admitted to ICU over a 6-month period, 12,850 (14.6%) patients died, with treatment limitations (withholding or withdrawing life-sustaining treatment) occurring frequently (80.9%). Common factors associated with treatment limitation included patient’s age and chronic disease, together with the presence of country-specific end-of-life legislation [20,21]. The decision to withdraw extensive supportive care is made by ICU physicians, based on measurable indicators of physical functioning and objective observations of vital signs. As most critically ill patients lack decision-making capacity and family members often serve as surrogate decision makers, decisions about the end of life should involve the family. Unfortunately, not only do data about true family participation in end-of-life decisions remain scarce, but end-of-life communication with families or surrogates varies markedly in different global regions; according to the Ethicus-2 study, discussion with family occurred only in 46.4% of cases in southern Europe, while in northern and central Europe percentages were significantly higher at 95.0% and 74.9.%, respectively [22,23]. For Greece, data are even more limited. In a national Greek study conducted across 18 multidisciplinary Greek ICUs dating back to 2015, 71.4% of 149 doctors and 59.8% of 320 nurses responded that families were not actively involved in discussion of life-sustaining treatment, confirming that in Greece fear of litigation is still considered a major barrier to properly informing the patients’ relatives about end-of-life decisions [24]. Since no clear, discontinuation criteria are defined in other countries either, the final decision always lies with the physicians, and is usually based on their experience and training [25]. However, making life-or-death decisions for another person is never an inconsequential decision for physicians, as reflected in increased burnout and distress among medical staff [26] and less empathetic and more cynical behaviour towards ICU patients [27]. In this regard, the present study suggests that the psychological pressure faced by medical staff does not affect their assessment of whether patients meet the withdrawal criteria. Poor quality of life in future and futility of treatment were found to be the most important criteria for the discontinuation of life-sustaining measures, while patients’ gender or age as well as treatment costs did not significantly influence the decision. It was also found that ICU physician assessment accurately assesses patients’ chances of survival during their hospital stay and in the immediate future. It is worth mentioning that in the Ethicus-2 study, 20% of patients with treatment limitations eventually survived the hospital stay, and the percentage in the earlier Ethicus-1 study (1999–2000) was even lower [20,21].

On a different note, in the burdensome environment of an ICU, members of the patient’s family are in a vulnerable position where depressive symptoms [28,29] and higher risk of anxiety and stress-related disorders are commonly reported [30]. Given this great psychological pressure, it is not surprising that for out of ten family members did not realistically assess the health progression of their relatives in ICU. In particular, the fact that the psychological symptoms of the family members were independent of the severity of the patient’s condition is supported by the findings of previous research [31]. 

Nevertheless, a realistic view is always necessary, because unjustified optimism makes it difficult to adjust to the loss of a loved one, promotes feelings of meaninglessness, and can lead to painful after-experiences in the ICU. This was demonstrated in the study by Sjökvist et al. [15], where the general public stated that they preferred greater influence from patients and families compared with intensive-care physicians in decisions to withdraw life support. The results of the present study further clarify these differences by highlighting mental resilience as the trait that determines the consent of family members to intensivists’ assessments. That is, this study further demonstrates the importance of resilience, previously shown in the study by Stamou et al. [13], as a psychological trait that reduces the overall psychological burden of ICU attendants. 

Moreover, resilience, as commonly defined as the process of adapting well in the face of adversity, trauma, tragedy, threats, or significant sources of stress, is highlighted as a key feature facilitating the transition from the initial emotional distress experienced by family members in the ICU to a sense of regained control [13]. As one such key feature, resilience provides family members with the right conditions to seek and create meaning in their situation and gives them purpose in contributing to their relative’s recovery. Therefore, an appropriate collaborative approach should be developed between family members and healthcare professionals, to address the patient’s needs while providing emotional and psychosocial support to their families [32]. Specifically, it is suggested that initiatives aiming to strengthen mental resilience will help relatives’ agreement with the opinions of critical care physicians, enhance quality of communication, reduce feelings of frustration and dissatisfaction from intensivists as well as relatives, and improve the overall satisfaction of the patient’s companions about the care their loved one receives [33,34].

After hospitalization in the ICU, most surviving patients require constant and long-term care and are unable to care for themselves. This situation, commonly referred to as post-intensive care syndrome (PICS), can affect the patient’s body, thoughts, feelings, and mental state [35]. Of course, this also puts strain on the family environment, both psychologically and financially. In the context of the two theoretical scenarios, caregivers of a surviving patient were reluctant to attribute family responsibility for life-sustaining decisions as their patients aged, a finding that indicates that their personal experiences strongly influenced the responses to the theoretical scenarios. It might be argued that positive health progress appears to predispose respondents to family involvement, while poor progress or the death of a loved one appears to reduce the desire to involve the family in life-support decisions affecting the ICU patient. Overall, a biased attitude was evident in our study, with personal experiences strongly influencing responses to theoretical scenarios. We found that the death of the patient distances the caregiving relative from the traumatic event of their family member’s hospitalization in the ICU. Furthermore, results indicated a limited validity of these instruments as decision-making aids with regard to the involvement of the family in withdrawal decisions in the ICU environment. 

There is ample evidence in the current literature that family members desire a more active role in end-of-life decision-making, in order to communicate patient’s wishes [36]. There is also consensus that end-of-life decisions should be viewed as shared decisions, with shared responsibility between the care team, the patient, and the family [37]. In this context, it is of paramount importance for intensivists to provide patients and their families with reliable information to help them decide whether withdrawal of life-support measures is the appropriate medical option [38]. Complete and comprehensible information about the medical data supporting the discontinuation of patient support could help loved ones to resolve their doubts about the treatment being offered, to understand the futility of the treatment, and to appreciate the severely reduced quality of life that awaits the patient if they survive after ICU [39]. Although the crucial need for complete and accurate information for family members of ICU patients has been widely reported [40,41] this factor seems to be underestimated by caregivers [42]. Since treatment futility is a rather vague concept and various attempts have been made to resolve this problem [43,44,45], there is a need for improvement in the communication skills of ICU staff so they can better distinguish and describe to relatives the individual aspects of treatment futility. With regard to the goal of family consent to the doctors’ decisions, an additional initiator could be the family’s right to additional patient care, which has also been described in the literature as a demand [44]. This opportunity would allow family members to experience the situation, recognize the futility of treatment, and create personal meaning for the potential loss of their loved one, making the loss gentler for them. Especially in the current context of COVID-19 with the noted shortage of ICU beds worldwide, better communication skills and methods between family members and caregivers will enable faster decisions and allow medical staff to provide medical care and hospitalization to people who need it most [45,46].

To the best of our knowledge, this is the first study to examine personal experience and resilience in relation to family members’ realistic views of ICU patient health as factors influencing attitudes towards end-of-life decisions.The use of disease severity scores such as GSC, APACHE, and SAPS to compare against family views is one of this study’s strengths.

Our study had some limitations. First, our results reported responses from one single ICU; hence their lack of generalizability should not be ignored. Second, family members who declined to participate at the second phase of the study i.e., six months after the ICU experience, may have reported different attitudes and perceptions. Furthermore, we did not assess post-ICU distress symptoms or post-ICU resilience and are therefore unable to justify the role played by such factors in long-term attitudes towards end-of-life decisions. Finally, since the data for the hypothetical scenarios were collected six months after ICU experience, the possibility of recall bias cannot be ruled out.

## 5. Conclusions

Family members of patients admitted to ICU have increased needs in terms of assurance, proximity, and information, and these requirements should be carefully considered by ICU staff [2]. The results of our study indicate that resilience as a personality trait was associated with an increased likelihood of agreement between family members and physicians’ perceptions of the patient’s health. Thus, it is suggested that mental resilience initiatives can help family members to adapt well to the overwhelming experiences in ICU and to recognize the situation pragmatically. In particular, the development of a philosophy of family-centred care should be a priority, with formal assessment of families taking place shortly after admission, followed by development of an appropriate care plan [47]. From this perspective, a collaborative approach between family members and medical staff will enhance the quality of communication, reduce feelings of frustration and dissatisfaction among physicians and relatives, and improve overall satisfaction with the care received.

Furthermore, family members’ perceptions of the patient’s health progress are related to their psychological characteristics, while the way they responded to the two hypothetical scenarios was related to their patient’s health progress (Appendix A). Therefore, it is cautioned that family members may have difficulty separating what they feel is best from what they believe the patient would think best [48]. It is suggested that the involvement of family members in important decisions regarding the patient life-support should be required of physicians working in critical care, while it appears that the ultimate decision should remain the sole responsibility of medical staff. However, we believe our findings merit further investigation with increased consideration given to the communication skills between ICU staff and family members; a factor that we did not examine herein. 

Finally, the appropriateness of hypothetical scenarios for ascertaining citizens’ perceptions is not supported, or at least their more cautious use and careful interpretation of the responses is suggested, which should consider the respondents’ recent exposure to relevant traumatic events as well as the trajectory of these events.

## Figures and Tables

**Figure 1 healthcare-11-00345-f001:**
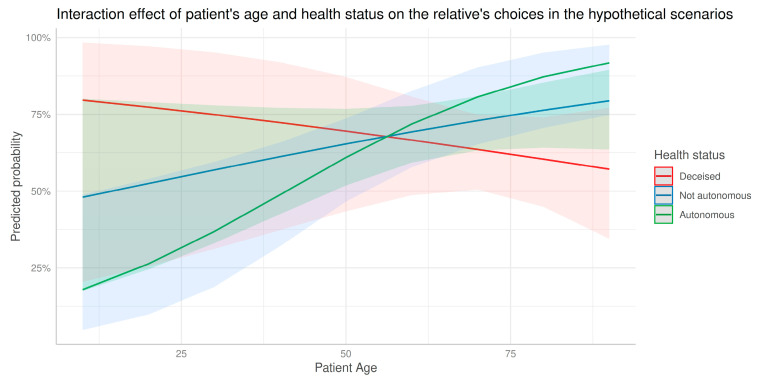
Interaction effect of age and health status on family members’ selections in the hypothetical scenarios. Predicted probability refers to the probability of a response that the patient himself or the family should be involved in the decision-making.

**Table 1 healthcare-11-00345-t001:** Family members’ and patients’ characteristics.

Family Members’ Characteristics (*N* = 189)	Mean (SD)
Age	46.5 (11.4)
Gender	Frequency (%)
Women	111 (58.7%)
Men	78 (41.3%)
Type of relation	
Spouse/partner	36 (19%)
Child	97 (51.3%)
Parent	14 (7.4%)
Other	42 (22.2%)
Stay with the patient	77 (40.7%)
Patients’ characteristics (n = 162)	Mean (SD)
Age	64.4 (17.2)
Gender	Frequency (%)
Women	52 (32.1%)
Men	110 (67.8%)

*N* = sample of the family members, n = sample of patients admitted to ICU.

**Table 2 healthcare-11-00345-t002:** Intensivists’ criteria for withdrawal or non-escalation of support measures.

Criteria ^(1)^	Frequency (%) ^(2)^	The Patient Meets the Withdrawal Criteria ^(3)^
Lack of future quality of life	49 (30.2%)	38 (82.6%)
Prolonged lack of quality of life	41 (25.3%)	34 (73.9%)
Futility of treatment	27 (16.7%)	25 (54.3%)
Body pain	19 (11.7%)	19 (41.3%)
Wishes of relatives	9 (5.6%)	9 (19.6%)
Moral pain	6 (3.7%)	6 (13%)
Patient’s age	6 (3.7%)	5 (10.9%)
Cost	0 (0%)	0 (0%)

^(1)^ Descending frequency order; ^(2)^ Percentage of the total *n* = 162 patients; ^(3)^ Percentage of the 46 patients who were judged to meet the withdrawal criteria.

**Table 3 healthcare-11-00345-t003:** Patients’ survival at ICU discharge and 6 months later, and survival prediction.

Time	Patients (n)	Survival	The Patient Met the Withdrawal Criteria		Intensivists’ Survival Prediction Indexes		
					Percentageaccuracy ^(1)^	Sensitivity ^(2)^	Specificity ^(3)^
			No	Yes			
Discharge from ICU	162	119	98	21	75.9%	84.5%	54.3%
After six months	112/119 *	87	78	9	71.0%	67.2%	80.4%

^(1)^ Percentage of patients correctly classified as survivals or non-survivals. ^(2)^ Percentage of patients classified as not meeting the survival criteria who subsequently survived. ^(3)^ Percent of patients classified as meeting the withdrawal criteria who died. *n* = sample of patients admitted to ICU. * 6 months after ICU discharge, it was possible to contact family members for 112 out of 119 patients.

**Table 4 healthcare-11-00345-t004:** Family members’ assessment of the progress of patient health and comparison to the intensivists’ assessment.

Family Member’s Assessment about Patient’s Health Progression	N (%)	The Corresponding Patient Meets the Withdrawal Criteria (Intensivist’s Judgment)		Agreement of Judgment (Family Member’s Judgment Compared to Intensivist’s)		
		No	Yes	Agree ^(1)^	Not Agree	
					Pessimistic	Optimistic
Hopeless	14 (7.4%)	7	7	7	7	
2	12 (6.3%)	9	3	3	9	
3	29 (15.3%)	22	7		22	7
4	41(21.7%)	29	12	29		12
Hopeful	86 (45.5%)	69	17	69		17
Total	182 (100%)	136	46	108	38	36

^(1)^ Family member’s judgment compared to intensivist’s. *N* = sample of the family members.

**Table 5 healthcare-11-00345-t005:** Logistic prediction model of agreement between attendants’ and intensivists’ assessments.

Variable	B	SE	Wald	df	p	Exp B	95% C.I	
							Lower	Upper
Intercept	1.005	1.508	0.444	1	0.505	2.731		
Patient’s demographic								
Gender	−0.293	0.354	0.683	1	0.408	0.746	0.373	1.494
**Age**	−0.020	0.010	3.843	1	0.050	0.980	0.960	1.000
Family members’ data								
Close relation	−0.424	0.482	0.773	1	0.379	0.654	0.254	1.684
Gender	0.328	0.337	0.948	1	0.330	1.388	0.717	2.688
Age	−0.023	0.015	2.465	1	0.116	0.977	0.949	1.006
Living with the patient	−0.384	0.355	1.170	1	0.279	0.681	0.340	1.366
**Resilience**	0.031	0.014	5.268	1	0.022	1.032	1.005	1.060
Self-compassion	−0.044	0.373	0.014	1	0.906	0.957	0.461	1.987

**Table 6 healthcare-11-00345-t006:** The hypothetical clinical scenarios.

Scenario 1	Scenario 2
A 60-year-old married woman with severe cancer and pneumonia needs the assistance of a ventilator in order to breathe. The woman will die within 24 h if the ventilator is withdrawn. The woman’s physician is completely convinced that she will die within a period of 1 month regardless of what treatment she receives. The woman is exhausted by her severe disease but fully conscious and able to express her wishes. The intensivists are considering withdrawing the ventilator and allowing her to die, so she will no longer have to suffer.	A 65-year-old married man was in a serious accident in which he suffered head injuries. One month later he is still unconscious and needs the assistance of a ventilator in order to breathe. The man will die within 24 h if the ventilator is withdrawn. The physician is completely convinced that he will not wake up, although he might live for a while if the ventilator is kept in place. The intensivists are considering withdrawing the ventilator treatment and allowing him to die.
***Question 1***: The intensivistsraise the question of continued ventilator treatment. Who should participate in this discussion? Answers: The patient, *n* = 102 (66.7%) The family, *n* = 82 (53.6%) Only the intensivists, *n* = 13 (8.5%) Uncertain, *n* = 12 (7.8%)	***Question 1***: The intensivists raise the question of continued ventilator treatment. Who should participate in this discussion? Answers: The family, *n*= 133 (86.9%) Only the intensivists, *n* = 17 (11.1%) Uncertain, *n* = 20 (13.1%)
***Question 2***: Assuming that the intensivists have brought up the question of ventilator treatment for discussion, whom do you believe should decide whether or not the ventilator treatment should be continued? Answers: The patient, *n* = 106 (69.3%) The family, *n*= 78 (51.0%) Only the intensivist, *n* = 78 (51.0%) Uncertain, *n* = 14 (9.2%)	***Question 2***: Assuming that the intensivists have brought up the question of ventilator treatment for discussion with the family, whom do you believe should decide whether or not the ventilator treatment should be continued? Answers: The family, *n* = 109 (71.2%) Only the intensivist, *n* = 89 (58.2%) Uncertain, *n* = 21 (13.7%) The treatment should not be stopped, *n* = 12 (7.8%)

*n* = groups of answers within the total sample.

**Table 7 healthcare-11-00345-t007:** Effects of patients’ characteristics on family/patient selection in the two theoretical scenarios.

Variable	B	SE	Wald	df	p	Exp B	95% C.I	
							Lower	Upper
Intercept	−0.075	0.972	0.006	1	0.939	0.928		
Agree with intensivists during hospitalization	0.584	0.393	2.215	1	0.137	1.794	0.831	3.872
Health Condition			5.118	2	0.077			
Not autonomous vs. Deceased	−4.892	2.769	3.122	1	0.077	0.008	0.000	1.707
**Autonomous vs. Deceased**	−5.319	2.360	5.079	1	0.024	0.005	0.000	0.500
Age	0.015	0.015	0.988	1	0.320	1.015	0.986	1.044
**Age × Condition**			6.873	2	0.032			
Not autonomous vs. Deceased	0.068	0.041	2.820	1	0.093	1.071	0.989	1.159
**Autonomous vs. Deceased**	0.088	0.034	6.864	1	0.009	1.092	1.022	1.166

## Data Availability

Not applicable.

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
