# Peer review of "Agreement between Family Members and the Physician’s View in the ICU Environment: Personal Experience as a Factor Influencing Attitudes towards Corresponding Hypothetical Situations"

_healthcare, 2023, doi:10.3390/healthcare11030345_

Round 1
Reviewer 1 Report
This is a very interesting study investigating a hot issue in daily clinical practice among ICU physicians and their relationship in terms of communication with patients' family members. Methods are robust and results are clearly and well presented. My only suggestion is that the authors should discuss in more detail other aspects of withdrowing decision-making, based on ETHICUS studies, as well as particular characteristics of family members. For instance, religious beliefs, socioeconomic status and geography have been found to affect significantly families' perception of end-of-life decisions, In addition, legislation differences among different countries might be a significant factor affecting such decisions, particularly in Greece.
Author Response
Review Report (Reviewer 1)
Comments and Suggestions for Authors
This is a very interesting study investigating a hot issue in daily clinical practice among ICU physicians and their relationship in terms of communication with patients' family members. Methods are robust and results are clearly and well presented. My only suggestion is that the authors should discuss in more detail other aspects of withdrowing decision-making, based on ETHICUS studies, as well as particular characteristics of family members. For instance, religious beliefs, socioeconomic status and geography have been found to affect significantly families' perception of end-of-life decisions. In addition, legislation differences among different countries might be a significant factor affecting such decisions, particularly in Greece.
- We thank the reviewer.
- Τo the best of our knowledge, Ethicus studies did not publish results regarding particular characteristics of family members. Data collection included only physicians' and patients' characteristics. We agree with your opinion that religious beliefs, socioeconomic status and geography have been found to affect significantly families' perception of end-of-life decision. We did not discuss this topic in detail, as the main aim of our study was to focus entirely on family members’ mental resilience as a personality trait and not to other characteristics such as religion, social rank and ancestry or descent.
- However, we have now added in the discussion the following statement “For example, in the Ethicus-2 study, a prospective observational study of 199 ICUs in 36 countries, among 87,951 patients who were admitted to ICU over a 6-months peri-od, 12,850 (14.6%) patients died, with treatment limitations (withholding or with-drawing life-sustaining treatment) occurring frequently (80.9%); common factors as-sociated with treatment limitation included patient’s age and chronic disease together with the presence of country end-of-life legislation ” with additional references, page 8, lines 251-256.
- We have also added that “Since most critically ill patients lack decision- making capacity and family members often serve as surrogate decision makers, decisions about end-of-life should involve the family. Unfortunately, not only data about true family participation in end-of-life decisions re-main still scarce, but end-of-life communication with families or surrogates varies markedly in different global regions; in Ethicus-2 study, in southern Europe, limitations dis-cussed with family occurred only in 46.4% of cases while in northern and central Europe percentages were significantly higher, 95.0% and 74.9.% respectively. For Greece, data is even more limited. In a national Greek study held in 18 multidisciplinary Greek ICUs back to 2015, 71.4% of 149 doctors and 59.8% of 320 nurses responded that the family was not actively involved in life-sustaining treatment discussion, confirming that in Greece fear of litigation still considered a major barrier to properly informing the patients’ relatives about end-of-life decisions….” with additional references pages 8- 9, lines 259-269.
- Moreover, we added that “It is worth mentioning that in the Ethicus-2 study, 20% of patients with treatment limitations eventually survived the hospital stay, while the percentage in the earlier Ethicus 1 study (1999-2000) was even lower ”, with additional references, page 9, 282-284.
Reviewer 2 Report
Dear authors,
Thank you for submitting this paper. Overall, it is an interesting topic that may add value to the scientific audience. However, I have some major concerns about it that needs to be addressed and revised to assess the manuscript for publication.
Background:
The background try to introduce previous research of family members and their impressions from the ICU. However, I suggest that a deeper literature review of the role of family within the ICU overall is first introduced, and then to step forward to how family members assess their hospitalized family members condition.
The sentence “In fact, an unrealistic perception of the patient's condition by the relatives leads to feelings of discomfort in the decision to withdraw life support and to tensions that are reflected in overall satisfaction with health care system services” does not have a reference, and it needs to be one to confirm this statement.
The sentence: “little is known about factors influencing the decision to withdraw life support made by families” is unclear; do you mean how family members themselves experience that the clinician make such decisions, or do you mean if the family wants the treatment to end?
Then you introduce two major concepts; resilience and self-compassion. However, the background does not mention these and how they (maybe) relate to the family members within the ICU, why I don´t really get why these are of interest in the aim? These two concepts have many, may definitions and you need to state which you have used and describe its relation to family members within the ICU care.
Methods:
Please state the study design first, by a separate paragraph, before introducing a study sample.
Table 2 is a result, not a method.
Please introduce the instruments used in a separate paragraph, and make sure that the presentation of how, when and with who each data collection was made.
Results:
I find the results a bit hard to understand; please try to structure your results and tables. I think this has to do with the presentation of methodology; what was the max and min for each instrument and how was for example “resilience” measured; what is the cut off value of the resilience or self-compassion scale? Therefore, it is very hard to see if they were considered as “resilience” (deepening on what version of view of this concept you choose in the background) or not.
Please note how N and n are use din the results and Tables. Use N for the total study sample and n for sub-stuy samples or groups within the sample.
Figure 1 makes no sense to me, that needs to be clarified and justified how it is important for the aim of the study.
Discussion:
I miss a critical reflection of the results in relation to theoretical and psychological processes of both self-compassion, resilience and health efficacy, that are central components in the study.
Conclusion:
As I understand, you conclusion is that if family members do understand the view of the clinicians opinion on the health status if the ICU patient, they will be more resilient? However, you need to define what you mean with resilience in order to be able to draw such conclusions. Also, I don’t understand how an experience or resilience my be “biases” by the perception of something. Is that not part of the resilience? I would say that of course their answers on scenario questions are influenced by their own experiences- would you expect otherwise?
I suggest that you clarify exactly what the outcome of the results is, and how it should be use din ICU.
Overall, minor academic English language editions is needed.
Author Response
Reviewer comments and our response
We thank the reviewers for all the very insightful comments and useful directions. All changes are underlined in yellow throughout the text.
Review Report (Reviewer 2)
Dear authors,
Thank you for submitting this paper. Overall, it is an interesting topic that may add value to the scientific audience. However, I have some major concerns about it that needs to be addressed and revised to assess the manuscript for publication.
- We thank the reviewer for your valuable comments, which we tied to address below.
Background:
The background try to introduce previous research of family members and their impressions from the ICU. However, I suggest that a deeper literature review of the role of family within the ICU overall is first introduced, and then to step forward to how family members assess their hospitalized family members condition.
We have now added that “Additionally, the generally unexpected admission of a patient to the ICU can be particularly frightening and distressing for their loved ones.Higher levels of anxiety, depression and stress are commonly reported in the literature during admission while post-traumatic stress disorder and complicated grief occur after discharge. Nonetheless, family members are seen as an integral part of the healthcare process and the need for good collaboration is always questioned, with additional references, page 2, lines 43-49.
The sentence “In fact, an unrealistic perception of the patient's condition by the relatives leads to feelings of discomfort in the decision to withdraw life support and to tensions that are reflected in overall satisfaction with health care system services” does not have a reference, and it needs to be one to confirm this statement.
- We have now added the following three reference for this statement, please see page 2 line 65
- Halain, A., Tang, L. Y., Chong, M. C., Ibrahim, N. A., & Abdullah, K. L. (2022). Psychological distress among the family members of Intensive Care Unit (ICU) patients: A scoping review. Journal of Clinical Nursing, 31, 497– 507. https://doi.org/10.1111/jocn.15962
- McAdam, J. L., & Puntillo, K. (2009). Symptoms experienced by family members of patients in intensive care units. American journal of critical care : an official publication, American Association of Critical-Care Nurses, 18(3), 200–210. https://doi.org/10.4037/ajcc2009252
- Myhren, H., Ekeberg, Ø., & Stokland, O. (2011). Satisfaction with communication in ICU patients and relatives: Comparisons with medical staffs’ expectations and the relationship with psychological distress. Patient Education and Counseling, 85(2), 237–244. https://doi.org/10.1016/J.PEC.2010.11.005
The sentence: “little is known about factors influencing the decision to withdraw life support made by families” is unclear; do you mean how family members themselves experience that the clinician make such decisions, or do you mean if the family wants the treatment to end?
- As we have previously stated on page, 2 lines “The role of family members is very important, as they are called upon not to express their own opinion on whether the patient's life should be further mechanically assisted, but the patient's own view, based on their perception of the patient's personality and character or after relevant discussions with the patient prior to admission to the ICU”, we mean if the family wants the treatment to end based on relevant discussions with the patient prior to admission to the ICU.
- We have now added that “Moreover, little is known about the factors that influence the patient's family members when asked to make end-of-life decisions on the patient's behalf” page , line 68.
Then you introduce two major concepts: resilience and self-compassion. However, the background does not mention these and how they (maybe) relate to the family members within the ICU, why I don´t really get why these are of interest in the aim? These two concepts have many, may definitions and you need to state which you have used and describe its relation to family members within the ICU care.
- We have now added that “In this context, the concepts of mental resilience as the subject's ability to succeed despite the adversities they face in life , and self-compassion as the subject's ability to have a warm, caring, empathetic, and non-judgmental orientation versus the self in times of suffering and failure have been im-plicated as potential factors in the psychological well-being of families in the ICU”, with additional references, page 2, lines 68-73.
Methods:
Please state the study design first, by a separate paragraph, before introducing a study sample.
- Point taken! We have now added “A cross-sectional study was conducted to evaluate the psychological impact on relatives of critically ill patients using a quantitative methodology” page 3, lines 95-96.
Table 2 is a result, not a method.
- We have now moved the Table 1 in the results section with additional heading. Page 4, lines 157-163.
Please introduce the instruments used in a separate paragraph, and make sure that the presentation of how, when and with who each data collection was made.
- We have now added that “The CD-RISC consists of 25 items that are answered on a 5-point frequency scale (0 to 4). CD-RISC's total score ranges from 0 to 100, with higher scores indicating greater perceived resilience. The SCS consists of 26 questions answered on a 5-point frequency scale (1 to 5), and the total score is calculated as the overall mean after 13 of them are reversed. The total score reflected self-compassion as defined as a dynamic balance between the compassionate and uncompassionate ways in which individuals respond emotionally to pain and failure, cognitively understand their predicament and pay attention to suffering”, page 3, lines 119-127.
Results:
I find the results a bit hard to understand; please try to structure your results and tables. I think this has to do with the presentation of methodology; what was the max and min for each instrument and how was for example “resilience” measured; what is the cut off value of the resilience or self-compassion scale? Therefore, it is very hard to see if they were considered as “resilience” (deepening on what version of view of this concept you choose in the background) or not.
- We have now added that” The mean resilience score for the total sample of family members was 70.8 (SD=14.4) analogous to the general Greek population (MP=70.2, SD=11.4). The mean score for self-compassion was 3.3 (SD=0.5), a score that is considered moderate (2.5 to 3.5) due to the lack of clinical norms or scores to suggest that a person has high or low self-compassion”, with additional reference, page 6, lines 204-208.
Please note how N and n are use din the results and Tables. Use N for the total study sample and n for sub-stuy samples or groups within the sample.
- For all tables, we have now used N for the total sample of family members and n for the sample of ICU patients or groups within the sample.
Figure 1 makes no sense to me, that needs to be clarified and justified how it is important for the aim of the study.
- We agree with reviewer, and in this revised version we have omitted the figure 1. In this version, Figure 1 corresponds to the old Figure 2.
Discussion:
I miss a critical reflection of the results in relation to theoretical and psychological processes of both self-compassion, resilience and health efficacy, that are central components in the study.
- We have now added that ..“Moreover, resilience, as commonly defined as the process of adapting well in the face of adversity, trauma, tragedy, threats, or significant sources of stress, is high-lighted as a key feature facilitating the transition from the initial emotional distress experienced by family members in the ICU to a sense of regained control. As one such key feature resilience provides to family members with the right conditions to seek and create meaning in their situation and give them purpose in contributing to their relative's recovery. Therefore, an appropriate collaborative approach should be developed between family members and healthcare professionals to address the patient's needs while providing emotional and psychosocial support to their families”, page 10, lines 305-313.
Conclusion:
As I understand, you conclusion is that if family members do understand the view of the clinician’s opinion on the health status if the ICU patient, they will be more resilient? However, you need to define what you mean with resilience in order to be able to draw such conclusions. Also, I don’t understand how an experience or resilience my be “biases” by the perception of something. Is that not part of the resilience? I would say that of course their answers on scenario questions are influenced by their own experiences- would you expect otherwise? I suggest that you clarify exactly what the outcome of the results is, and how it should be use din ICU.
- We have now elaborated the issues raised by the reviewer by adding the following statements “Family members of patients admitted to the ICU have increased needs in terms of assurance, proximity, and information that should be carefully considered by ICU staff” page 11, lines 378-379. And on page 11, lines 382-390 we added that “Thus, it is suggested that mental resilience initiatives help them to adapt well to the overwhelming experiences in ICU and to recognize the situation pragmatically. In particular, the development of a philosophy of family-centred care appears to be a priority, with a formal assessment of families taking place shortly after admission and an appropriate care plan. From this perspective, a collaborative approach between family members and medical staff will enhance the quality of communication, reduce feelings of frustration and dissatisfaction among both physicians and relatives, and improve overall satisfaction with the care received.
- And on page 11, lines 391-395 we added that “Furthermore, family members' perceptions of the patient's health progress are related to their psychological characteristics, while the way they respond to the two hypothetical scenarios is related to their patient's health progress. Therefore, it is cautioned that family members may have difficulty separating what they feel is best from what they believe the patient would think is best.
- We have also modified the statement about biased conclusions as follows… However, we believe our findings merit further investigation considering the communication skills between ICU staff and family members; a factor that we did not examine herein. Page 12, lines 395-397.
Overall, minor academic English language editions is needed.
- We have now corrected minor academic English language issues
Round 2
Reviewer 2 Report
Thank you for your efforts to improve the paper. I have no further suggestions and I hope that you will continue to research this important question.